# Remotely Powered Two-Wire Cooperative Sensors for Biopotential Imaging Wearables

**DOI:** 10.3390/s22218219

**Published:** 2022-10-27

**Authors:** Olivier Chételat, Michaël Rapin, Benjamin Bonnal, André Fivaz, Josias Wacker, Benjamin Sporrer

**Affiliations:** 1CSEM, Electronics/Systems/Digital Health, Jaquet-Droz 1, 2002 Neuchâtel, Switzerland; 2CSEM, Integrated & Wireless Systems/System-on-Chip/ASIC for the Edge, Technopark, Technoparkstrasse 1, 8005 Zürich, Switzerland

**Keywords:** biopotential imaging, body surface potential, active electrode, dry electrode, cooperative sensor, wearables, medical device

## Abstract

Biopotential imaging (e.g., ECGi, EEGi, EMGi) processes multiple potential signals, each requiring an electrode applied to the body’s skin. Conventional approaches based on individual wiring of each electrode are not suitable for wearable systems. Cooperative sensors solve the wiring problem since they consist of active (dry) electrodes connected by a two-wire parallel bus that can be implemented, for example, as a textile spacer with both sides made conductive. As a result, the cumbersome wiring of the classical star arrangement is replaced by a seamless solution. Previous work has shown that potential reference, current return, synchronization, and data transfer functions can all be implemented on a two-wire parallel bus while keeping the noise of the measured biopotentials within the limits specified by medical standards. We present the addition of the power supply function to the two-wire bus. Two approaches are discussed. One of them has been implemented with commercially available components and the other with an ASIC. Initial experimental results show that both approaches are feasible, but the ASIC approach better addresses medical safety concerns and offers other advantages, such as lower power consumption, more sensors on the two-wire bus, and smaller size.

## 1. Introduction

The technology of *biopotential* measurements has been known for decades [1] and is widely used to measure ECG (electrocardiogram), EEG (electroencephalogram), EMG (electromyogram), etc. Today, most products are still based on adhesive *gel electrodes* applied to specific areas of the body’s skin and connected in a star arrangement to a *central unit* (e.g., a recorder or monitor) by shielded cables. Another solution, less commercialized but scientifically well known, features *active electrodes* [2]. Active electrodes have amplification electronics at the electrode to achieve a high input impedance and low output impedance. Although they do not require shielded cables, each electrode still requires at least two wires to be connected to the central unit in a star arrangement [3].

Implementing biopotential sensors in *medical wearables* is difficult for a few fundamental reasons. One of them results from the preferential use of *dry electrodes*. For dry electrodes to provide good quality signals, a higher input impedance of the amplification circuit (cables and amplifier) is required [4,5,6]. Active electrodes are a good solution to meet this requirement, but they must be powered. However, powering them from the central unit via their leads is not trivial in the context of medical devices. The basic safety standards [7,8,9] for medical electric equipment (words in small capitals have a defined meaning in the standards) require a maximum patient leakage current of 10 μA d.c. (or 10 mV across the 1 kΩ taken by the standards for skin) for ECG medical devices. In addition, the standards specify two means of patient protection [7]. Reliable double insulation of conductive tracks in wearables is difficult to achieve, especially due to the possible presence of *body fluid* (e.g., sweat, urine, etc.). To be considered as one mean, for working voltages lower than 60 V, solid insulation must pass the test of dielectric strength at a voltage of 500 V rms during 1 min under the worst expected condition (i.e., with body fluids and at the end of the expected service life) [7]. Therefore, for wearable devices, other means of patient protection than solid insulation, such as the electronic detection and active limitation of tiny patient leakage currents, are desirable.

Another reason that wearables are difficult to design, especially for devices requiring many electrodes (e.g., >100) such as for *electrographic imaging* (e.g., ECGi [10], EEGi [11], EMGi [12]), is the ‘star arrangement’ of conventional wiring approaches. In addition to the complexity of routing many cables, connecting them to the central unit where they meet results in expensive and cumbersome *connectors*, which is a major integration challenge for waterproof and reliable wearables. To solve these problems, an architecture where each electrode includes electronics to be interfaced in parallel with a bus is attractive because such an architecture is virtually independent of the number of electrodes. Buses with a small number of lines, such as one or two at most, are the most desirable, but more difficult because the bus must perform several functions (potential reference, current return, synchronization, communication, and power supply). *Cooperative sensors* [13,14,15,16,17,18,19], defined as active electrodes connected to a parallel bus of up to two wires, have been proposed to address these challenges, except for the power supply. Adding the power supply function—without interfering with the other functions and addressing the medical safety issue in wearables—is the main contribution of this paper.

Section 2 presents the previously known cooperative sensor technique characterized by a dedicated power supply (battery) per sensor and a bootstrap circuit allowing high input impedance. This section is essential to understand the functions that must remain unaltered despite the introduction in Section 3 of the power supply on the same wires. Two solutions are presented, one called ‘*Legacy approach with 500 Hz powering and off-the-shelf components*’ (because directly built on the cooperative sensors of Section 2) and the other named ‘*Approach addressing the safety issue with powering at 1 MHz and ASIC for 250 sensors*’. Section 4 provides implementation details and experimental results for both approaches. The paper ends with a conclusion (Section 5).

## 2. State-of-the-Art Cooperative Sensors

Active electrodes connected by *up to two wires* in a parallel bus arrangement are called *cooperative sensors*—‘cooperative’ because co-operation of at least two sensors, each measuring one biopotential, is required to obtain a difference of potential (such as how two fingers, at least, must cooperate to pick up a golf ball). Compared to conventional approaches [1,2,3,20,21], cooperative sensors benefit from a *parallel bus arrangement*, which contributes to the scalability of the system. Moreover, unlike the multi-wire bus of direct multiplexing [22], the complexity of their connection is reduced to a minimum (only two wires for all functions).

### 2.1. Basic Circuit and Their Interconnections

Figure 1 shows a patented generic mechanism [13,14] for synchronization and control of the cooperative sensors and for transferring the acquired signal to the central unit. To measure the voltage *e*, a *current electrode* (left in Figure 1), a *potential electrode* for potential reference (middle in Figure 1), and one potential electrode per independent biopotential channel (right in Figure 1) are required. The current electrode is often referred to in the classical technique [1] as *right-leg electrode* and its purpose is to provide a path for currents capacitively coupled to the electronics (central unit and cables) from the environment (e.g., 50 Hz or low-frequency currents resulting from motion in the earth’s electric field). The central unit sends synchronization information and other commands simultaneously to all cooperative sensors with the voltage source *U*. The received signal is picked up by the cooperative sensors with the voltage across their current source, i.e., the voltage between the two bus wires, and feeds the clock recovery and sync block S. The cooperative sensors use their current source to communicate their measured potential *v_i_* (or other signals) to the central unit by means of a modulation M. The central unit receives the composite information from all the sensors by detecting the current in one of the bus wires. The demodulation D allows to recover the individual signals. Note that there are many possible modulations M. Amplitude modulation of a frequency carrier is a simple example. Digitization combined with phase shift keying could be another. The dark gray box symbolizes the power supply (e.g., a battery) of the cooperative sensors.

Since the total gain of the acquisition chain (from the potential *v_i_* to the demodulated signal $v_{i}^{\prime }$) may be slightly different from sensor to sensor, the common-mode rejection when performing the difference $v_{i}^{\prime }-v_{j}^{\prime }$ might be insufficient. The problem can be neatly solved by the method described in [15], which relies on online identification of the $v_{i}^{\prime }/v_{i}$ transfer functions through an excitation common-mode voltage added to the voltage source driven by the *G* controller.

Figure 1 also shows that the two wires could be implemented as a textile spacer with both sides made conductive. Note that these conductive surfaces also implicitly act as ‘body shield’, in contrast to the classical approach where the shield must be explicitly added [23]. One side of the cooperative sensor is used as dry electrode and the other as bottom contact with the fabric. The upper contact is made on the other side of the fabric by the sensor attachment. Due to the integrated electronics (e.g., ASIC), the cooperative sensors can be very small (e.g., 4 × 4 mm^2^) while the dry electrode and the contacts to the fabric can be larger and flexible. As a result, the assembly of the sensors with the fabric is virtually cable-free and seamless, making it easier to meet the wearability constraints (flexibility, breathability, stretchability, washability).

The *G* controller ensures that the voltage between the lower wire and the body is approximately zero. As long as this voltage is always less than 10 mV for ECG and less than 100 mV for other biopotentials, e.g., EMG—which is the case—no insulation is needed, because the standards require ECG devices to be type cf (limit to 10 mV) and other devices type bf (limit to 100 mV) [7]. By default, the other side is not in contact with the skin and can only be touched intermittently, for example with the hand. This problem can be solved by electronic detection (see below). However, the detection of leakage currents will stop the operation of the device. Therefore, one may consider adding a layer of fabric to insulate the top conductive side in most situations and relying on electronic detection only for exceptional situations. Optionally, the outer side of this additional layer can be made conductive and connected to the skin to provide one mean of patient protection in the same idea as class i medical electrical equipment [7]. This conductive layer also shields the middle conductive layer with respect to EMC emission and immunity.

### 2.2. Floating Supply and Bootstrapping

Figure 2 shows in more detail an implementation of cooperative sensors [16,17,18] for biopotential measurement (the cooperative sensors described in [17,18] also measure bioimpedance for EIT, electrical impedance tomography, but in this paper, we focus only on biopotentials for simplicity). In this implementation, each cooperative sensor is powered by a battery. All batteries can be recharged simultaneously via the two-wire bus when the system is not worn. The safety issue related to the maximum leakage current of 10 μA d.c. is not applicable when the system is not worn by the patient—the connector to the charger is made so that it is impossible for the patient to wear the system during charging. Once in the sensor, the stored charges are prevented from leaving the sensor by diodes. Thus, there is no possible harm resulting from insulation failure, for example, due to body fluid. Note that communication is not a problem because at higher frequencies, higher leakage currents are allowed, and communication requires lower voltages (which reduces possible leakage currents).

The input impedance of the cooperative sensors is significantly increased by a power supply bootstrapping approach [19] that takes advantage of the degree of freedom provided by the floating power supplies of cooperative sensors. The measured potential connected to the negative input and the middle potential of the supply connected to the positive input are made equal by the operational amplifier. This bootstrapping strategy is preferred to other well-known solutions, such as positive feedback [24], because the performance is higher and the risk of instability is lower. The electrode and its connection to the operational amplifier are shielded with the middle potential of the power supply in a manner equivalent to the driven shielding of classical techniques.

## 3. Method

Cooperative sensors, with their very high input impedance suitable for dry electrodes (thanks to bootstrapping) and their connection via a 2-wire parallel bus, solve the integration difficulties related to wiring and the scalability issues of star-arrangement connections, thus paving the way for biopotential imaging wearables comprising many electrodes. Like any active electrode, cooperative sensors require power. Safety constraints resulting from poor and unreliable insulation in wearables due to wear and tear, as well as the possible presence of body fluids, make remote power from the central unit a real challenge for medical devices where the maximum leakage current is 10 μA d.c. [7]. The cooperative sensor approach in Figure 2 avoided this problem by using one battery in each sensor and implementing a recharging strategy using the 2-wire bus when the vest is not worn (e.g., when placed on a hanger modified as a charger). However, one battery per sensor is expensive, heavy, and cumbersome.

This section presents two solutions for remote powering of cooperative sensors. The first solution (Section 3.1) solves the difficulty of adding the power supply function to the bus without increasing the noise of the measured biopotentials beyond the limit required by the standards, and without interfering with the synchronization and bidirectional communication between the sensors and the central unit. The second one (Section 3.2) also aims to solve the safety issue mentioned above, significantly reduce the volume of the sensors (e.g., from 7.5 cm^3^ to 0.3 cm^3^) and the power consumption (e.g., from 5.8 mA to 150 µA), operate with more sensors (e.g., from 20 to 250), and implement a bootstrap that does not require floating batteries as in Figure 2.

### 3.1. Legacy Approach with 500 Hz Powering and Off-the-Shelf Components

Figure 3 shows the circuit in Figure 2 with modifications to allow for remote powering. The cooperative sensors ‘harvest’ their energy from the 2-wire bus powered by the voltage source *U* of the central unit. Capturing power from the bus is symbolized in the first cooperative sensor (middle of Figure 3) by a current source whose current is in phase with the voltage *U* (power is consumed when current and voltage have the same sign, as with a resistance).

The resistance of the bus wire is low but the supply current high, and the resulting voltage drop on the lower wire is in series with the potential *e* to be measured. Therefore, the supply and measurement must be in separate frequency bands to minimize interference with the measurement. The first-order bandwidth of biopotentials is, for instance, 0.05–150 Hz for ECG. Therefore, constant current sensors will give a disturbance spectrum theoretically at 0 Hz, but even if they are carefully designed with active control to keep the current constant, it is very difficult to avoid overlap with the ECG band. To avoid this interference and to easily obtain a bipolar supply in the sensors (see below), the voltage *U* was chosen as a square wave at 500 Hz corresponding to the Nyquist frequency of the 1 kHz sample rate. Interference with the fundamental (500 Hz) is first avoided by fine-tuning the phase of the supply square wave (at the Nyquist frequency, a sinewave has all its samples equal to zero; only the cosine has non-zero samples). Any remaining energy of the fundamental—in practice, it is impossible to have the same adjusted phase for all sensors—is removed with a first-order notch filter by signal processing (i.e., moving average of two consecutive samples). The harmonics (1 kHz, 1.5 kHz, 2 kHz, …) must be removed before sampling, otherwise they create aliases at 0 (even harmonics) and 500 Hz (odd harmonics). An ideal square wave has energy at the odd harmonics and the 500 Hz notch filter is, therefore, necessary. To remove the even harmonics from a non-ideal square wave (an unavoidable situation in practice due to the asymmetry of power consumption for positive and negative currents), a third-order delta-sigma analog-to-digital converter was chosen, because the digital antialiasing filter of a delta-sigma converter is a comb filter with notches at multiples of 1 kHz (sample frequency).

The harmonics of a 500 Hz power supply are also low enough in the MHz range that they do not significantly disrupt the digital communication (modified to 1.28 Mb/s, in both directions, to conform to the frequency required by the delta-sigma converter). The bits just past the edges of the 500 Hz supply are disturbed and, thus, removed from the communication payload (in our prototype, 110 bits are removed, i.e., 17%).

In Figure 2, the upstream channel (from the central unit to the sensors) is realized as a voltage source and the downstream channel (from the sensors to the central unit) as current sources. An alternative is to interleave the two channels, each with its own time slots. In this way, both the up and down channels can use voltages (or currents). In Figure 3, the current sources have a resistance in parallel. Thevenin’s equivalent, a voltage source with the resistance in series, is easier to implement since the voltage source is simply a digital output. The capacitance and inductance are chosen so that their resonance is at 2.56 MHz, and the resistance so that the RLC triplet implements the first-order bandpass filter used first to prevent the communication band from overlapping the biopotential band and second to avoid EMC problems that sharp edges in the digital signal can cause. Note that the RLC triplet is a parallel assembly of R, L, and C in both the central unit (voltage source *U* at 0) and in the sensors (the capacitance in series with the inductance is chosen so that its impedance is negligible at communication frequencies). The received signal is the voltage on the RLC triplet. After its reconstruction with a high-pass filter and Schmitt trigger, the digital signal is demodulated in the D-block to obtain $v_{i}^{\prime }$.

Since the LC blocks any current at the communication frequency, the voltage on the RLC triplet is the result of a voltage divider consisting of the emitter resistance and all other resistances (of the receivers) in parallel. The consequence is that the received voltage is the emitter voltage divided by the number of units (i.e., sensors and central unit). Therefore, this approach limits the number of units in practice to approximately 20.

The 500 Hz supply square wave is provided by the voltage source *U*, which is easily realized with switching transistors. The impedance of the central unit and sensor inductances is negligible for the supply current. Therefore, the supply square wave is rectified by diodes in the sensors to provide a positive and a negative voltage on the storage capacitors.

### 3.2. Approach Addressing the Safety Issue with Powering at 1 MHz and ASIC for 250 Sensors

The approach of Figure 3 solves the problem of remote power supply with respect to interference with biopotential measurement and communication on the 2-wire bus, but does not solve the safety problem for medical devices, because the allowed patient leakage current at 500 Hz is also 10 μA for ECG devices that must be type cf [8]. For other biopotentials, there is already an appreciable advantage over d.c., since the devices can be type bf and the allowed patient leakage current is ten times higher (i.e., 100 μA). Assuming a per sensor consumption of 8 mA (see next section) and 25 sensors, the supply current on the bus is 200 mA. Detecting a 100 μA leak from monitoring the supply current at the central unit is a difficult task (1 part in 2000). In addition, all sensors must have a buffer current source that makes closed-loop adjustments to its current to ensure that the current of any sensor is exactly 8 mA (again with high accuracy).

To better address the safety issue, including the ten-fold increase in ECG requirement, and to increase the number of sensors by a factor of 10, i.e., to up to 250, we need to reduce the power consumption of the sensors (say, by a factor of 20, i.e., to 400 μA) and move the power supply frequency to 1 MHz where the standards allow a patient leakage current of up to 10 mA (which is also the absolute maximum). This will make detecting a leakage current much easier (1 part in 10).

Note that the 8 mA and 400 μA mentioned above are the current of a sensor as measured in the bus. The sensor itself consumes half of this current, i.e., 4 mA and 200 μA, respectively. The factor of two comes from energy conservation, i.e., the bus supply voltage *U* is ±VCC/2 with a current of ±2*I* (rms value 2*I* for square waves) which allows sensors with the dual half-wave rectifier (assuming perfect diodes) to have a VCC supply and *I* current for the electronics.

To reduce power consumption by a factor of 20, we developed an ASIC (application-specific integrated circuit) that optimized each electronic function. In addition, we eliminated the digitization (analog-to-digital converter). The transmission of analogue values instead of bits has also increased the throughput (required for 10 times more sensors). The 1 MHz power supply is now interleaved with the communication, i.e., every other period the power supply is replaced by the communication [25]. Figure 4 shows the principle of this implementation. The inductances are no longer needed (which is good because they cannot be integrated into silicon as passive components). A switch reroutes the 1 MHz square-wave signal either to the rectifier diodes and storage capacitors (harvesting period) or to the communication current source. Before transmission, the biopotential is amplified and filtered. A high-pass filter prevents the transmission of the electrode offset that can be as high as 300 mV according to the standards, and thus improves the signal-to-noise ratio of the analogue communication. The M modulator simply selects the right time slot for the sensor to transfer its value. All sensors sample their biopotential at the same time. The value to be transmitted is stored in a capacitor until transmission.

Bootstrapping is achieved with the regulated supply rails VCCF and GNDF (specific for each sensor) following the electrode potential (with offsets). This is obtained by the follower controlling the reference of the LDO voltage regulators. Assuming an LDO gain *g* (i.e., the LDO outputs a current *i* = *gu* where *u* is the voltage error of the LDO output), the input impedance of the open loop circuit is magnified by *gz* at low frequencies. High gain at low frequencies can be achieved if *z* behaves like a capacitance at low frequency. At higher frequencies, for stability reasons, it is preferable for *z* to behave like a resistance. The open loop input impedance is essentially the input impedance of the follower (typically 10 pF). The bootstrap magnifies this impedance by *gz*, allowing the circuit to have a very high input impedance at low frequencies [25]. Compared to Figure 3 where bootstrapping is not implemented, this bootstrap also makes shielding of the sensor input more efficient and natural, as the ground and power rail planes provide implicit (driven) shielding.

This paper only describes the measurement of biopotentials. However, the developed ASIC is also capable of measuring bioimpedance (for EIT) and can be interfaced with an electret to pick up body sounds (stethoscope).

### 3.3. Comparison to Existing Work

Table 1 shows a comparison with existing work to further highlight the significance of the presented work. Compared to the closest state of the art, remotely powered cooperative sensors do not require a local power supply (e.g., a battery per sensor) which allows them to be miniaturized, among other things. Being able to monitor the leakage currents allows wearables to be safe (in the context of medical standards) without relying on the insulation of conductors in a garment and without the need for waterproof connectors.

## 4. Results

### 4.1. Legacy Approach with 500 Hz Powering and Off-the-Shelf Components

#### 4.1.1. ECG Study Prototype

Figure 5 shows a study prototype with 224 sensors for ECGi based on the approach shown in Figure 3 with a 500 Hz power supply and commercially available components. The device is cf and is defibrillation-proof between two electrodes (verified according to [8]). The central unit has recording and wireless communication capabilities and is powered by IEC 62,133 batteries. A close look at Figure 5a shows that the sensors are not connected on a single two-wire bus. This is because the approach in Figure 3, as described in the previous section, cannot have more than approximately 20 sensors on the same bus. Instead, columns of eight sensors are each connected by a two-wire bus to repeaters, which in turn are connected to a two-wire bus with 14 repeaters, and the system is doubled with left and right banks from the central unit. So, it is not a fully parallel bus topology but a tree arrangement comprising 2 × 14 × 8 = 224 sensors. The sensors are equipped with a CPLD (complex programmable logic device) to implement the PLL and internal clock reconstruction, a lossless compression scheme to reduce the 24 bits of delta sigma to 10 bits, and information exchange with the bus.

The study prototype passed all hardware performance tests [8], including tests related to defibrillator protection. The device was designed to have lower noise than required by ECG standards because its intended use is ECGi. The measured rms value of ECG noise is 2.5 μV and the peak-to-peak value 20 μV over 10 s. The recovery time after defibrillation is less than 300 ms (standards require a maximum of 5 s). The study prototype at this phase of development was not built to be worn. However, Figure 5b shows the signals measured on a subject with two pairs of electrodes on either side of the chest. One pair consisted of dry stainless-steel electrodes and the other of Ag/Ag^+^Cl^−^ gel electrodes. For both signals, noise is barely visible. Note that the two signals are not identical because they result from close but different electrode positions.

Figure 6 shows a simplified electronic schematic of the entire device (the safety and defibrillator protections are not shown) implementing the principle of Figure 3. The central unit (left) is connected to one of the 14 repeaters (middle) of the right bank, which in turn is connected to one of the eight cooperative sensors (right). The voltage source *U* of the central unit (see Figure 3) is implemented with switch transistors alternatively connecting the battery (4.6 V) or a bypass (0 V). The repeaters, cooperative sensors, and central unit harvest energy from the two-wire bus using dual half-wave rectifiers. A capacitor connected just before the half-wave rectifiers removes the offset of *U* (2.3 V) and provides a bipolar VCC/GND supply (±2.3 V) which is symmetrical with respect to REF, the potential of the lower line of the two-wire bus. The LCR triplet consists of an inductor, a parasitic capacitance and two resistors in parallel directly connected to digital outputs of a CPLD (Logic) operating in push-pull mode to send a signal. The edges of the received signal is regenerated by a Schmitt trigger.

In the implementation of Figure 6, to keep it simple, the central unit has two electrodes. Therefore, the *G* controller is analogue (the operational amplifier) since the *v*_1_ signal comes from the extra electrode of the central unit. The repeater receives the digital signal on one side and retransmits it on the other side. This is achieved at the cost of a one-bit delay, which is not a problem because the shift is taken into account by the communication protocol. The same communication principle as described in [18] is used. The sensor electrode is connected to the positive input of an operational amplifier. Two resistances define the gain of the first amplification stage and an RC circuit at the output of the operational amplifier implements the coarse antialiasing filter required by the delta-sigma converter.

To measure skin impedance (e.g., for a lead-off detection function), the RC circuit connected to the electrode provides a simple way to inject a tiny current of ±1 nA at 500 Hz (square wave) with a digital output. The phase is chosen to be 90° offset from the sample times (cosine), because a sinewave at the Nyquist frequency has all its samples equal to zero. The capacitance is chosen to have a corner frequency just below 500 Hz to maximize the input impedance at lower frequencies where the biopotential is measured. The voltage drop resulting from the injection of this current through the skin impedance is superimposed to the biopotential signal. However, it can be isolated by taking the difference of two consecutive samples and ‘demodulated’ by taking only the even samples (down sampling by two). This processing is conducted in the CPLD of each sensor before the data are transferred to the central unit. Taking the difference of two consecutive samples (filter with transfer function 1 − *z*^−1^) also allows to efficiently ‘compress’ the bipotential signal (for example, on the 24 bits of the delta-sigma converter, only 10 bits are enough to ‘encode’ a medical ECG). For the biopotential, the decompression is performed in the central unit by the inverse filter 1/(1 − *z*^−1^). This compression scheme is lossless (except at 0 Hz, but 0 Hz is outside the bandwidth of biopotential signals). The notch filter at 500 Hz (sum of two consecutive samples, i.e., the filter with the transfer function 1 + *z*^−1^) needed to remove the supply disturbance is also realized in the sensor CPLDs. Therefore, the central unit must apply this corresponding inverse filter, i.e., 1/(1 + *z*^−1^), to recover the skin-impedance signal. Again, this decompression is lossless, except for the frequency exactly at 500 Hz. Therefore, the 500 Hz current for measuring skin impedance is turned on and off every 1 s to make a differential measurement (corresponding in the frequency domain to a frequency line at 499 Hz that is not affected by the lossless compression/decompression process). The entire processing scheme is shown in Figure 7.

#### 4.1.2. EEG Study Prototype

Figure 8 shows another study prototype based on the development presented in Figure 5 but reworked to handle dry electrode EEG. The modifications are mainly a higher gain for lower noise (0.7 μV rms, 5.4 μV pp over 10 s), a narrower bandwidth (0.5–50 Hz, 1st order), and a device limited to eight sensors. The noise is within the limit (6 μV pp over 10 s) accepted by the standard [26]. The obtained power spectra are displayed on the right and show the expected changes in power at certain frequencies resulting from the closed eyes.

The implementation circuit is similar to that in Figure 6, but does not include the repeaters, and the central unit connects only a two-wire bus with eight cooperative sensors. Table 2 gives the measured noise on the prototype resulting from digital processing of the difference in potentials (sensor 1 as reference) in the EEG bandwidth of 0.5–50 Hz specified by the standard [26].

### 4.2. Approach Addressing the Safety Issue with Powering at 1 MHz and ASIC for 250 Sensors

#### 4.2.1. ASIC Architecture

The ASIC can be mainly divided into two sections, as shown in Figure 9. The first section, in red, provides the interface to the two-wire bus. At startup, the power management unit uses the power square wave generated by the central unit to turn on all internal power supplies. After this is accomplished, a delay-locked loop in the clock and timing recovery block listens for the sync marker in the 1 MHz square wave. The sync marker is a periodicity break that marks the beginning of a sequence of 1,000,000 periods, i.e., there is sync marker every 1 s (see the illustration in Figure 10). The power management harvests current from the bus only every other period. The other periods are used for communication. Only the ASIC whose ID corresponds to a given communication slot transmits its acquired sample to the central unit.

The second section of the ASIC, shown in green in Figure 9, provides the signal chains for a biopotential electrode and additional sensing functionality that can be integrated into a sensor. In the example in Figure 9, the additional functionality is a stethoscope. In addition, a current injection block is used to check the contact impedance of the electrode.

The biopotential acquisition chain starts with a unity gain buffer, as shown in Figure 9. The output of the reference buffer is used as a ‘floating reference’, that is, it is used as a ground reference for a positive supply rail and a negative supply rail that feed the reference buffer itself (see Figure 11). With this bootstrapping approach, the buffer supply perfectly follows the (a.c.) biopotential to be measured. As a result, the voltage on the parasitic capacitances at the input is asymptotically close to zero and, thus, virtually no current flows. This is equivalent to a drastic increase in the input impedance.

The vcc and vee supply rails in Figure 11 follow the floating reference determined by the buffer itself fed by vcc and vee. Positive feedback is possible and to avoid instability, the loop gain must be kept below unity. In the ASIC, this is achieved by a single-stage design that has a large power supply rejection ratio (PSRR) for vcc and vee within the bandwidths of the low dropout regulators (LDOs) generating these power supplies.

After the high input impedance follower, the ECG signal is chopped at a frequency of 1–12.5 kHz to avoid flicker noise during amplification. Then it is filtered and sampled before the transmitter sends the analog signal to the central unit via the bus.

#### 4.2.2. Central Unit Architecture

The central unit implementation based on the approach shown in Figure 4 is presented in Figure 12. The two wires A and B of the bus are powered by the voltage source *U*—a square wave at 1 MHz with a voltage of ±*V_h_*. Every other 1 MHz period, the sensors are not powered by the bus. Instead, one of them—as determined by its address—sends into the transimpedance capacitor, during the period when *U* = *V_h_*, a quantity of electrical charge proportional to the measured potential. The voltage at the terminals of this capacitor is then sampled by the ADC to be demodulated by the microcontroller. After sampling, the transimpedance capacitor is reset by shorting its terminals with a switch. The controller *G* in Figure 4 is implemented with a pass-through as in Figure 6.

#### 4.2.3. Implementation Results

Figure 13a shows the ASIC package mounted on a micro-PCB (7 × 7 mm^2^). On the other side of the PCB (not shown in Figure 13), there are some capacitors and other components that were not practical to be part of the ASIC. However, their footprint is very small, and since the chip itself is only approximately 2 × 2 mm^2^ and the number of pins can be reduced to ≤20, a final sensor implementation of 4 × 4 mm^2^ could be targeted. The first integration test setup is shown in Figure 13b. The central unit development board is on the left and is connected via the two-wire bus to a cooperative sensor (i.e., the ASIC). This setup acquired the signal of an ECG simulator applied between the sensor and GND. Figure 13c shows the acquired signal for an ECG of 2 mV (peak R).

Another important step demonstrating the feasibility of the proposed approach is shown in Figure 14 where two ASICs (see Figure 13a) were applied to stainless-steel dry electrodes on the body, as shown in Figure 14. Despite the use of an optocoupler (DLN-4SE), some 50 Hz problem remained, but at a low enough level to allow acquisition of a clear ECG (see Figure 14). The 50 Hz problem should be fully resolved when the central unit will have been made ‘wearable’ with a floating power supply and when a few other defects discovered during this exploration phase have been corrected.

## 5. Conclusions

The two developments presented in this paper have demonstrated the feasibility of cooperative sensors for medical imaging wearables for biopotentials. The main contribution is the disclosure of circuits that allow remote powering of cooperative sensors via their two-wire bus.

Cooperative sensors are ideal for biopotential imaging wearables because they can be deployed in large numbers (>100) without suffering from the wiring complexity of conventional star arrangements. In an initial development, we demonstrated that high-quality ECG signals can be acquired from up to 224 defibrillator-proof dry electrodes connected via a two-wire bus (or rather, a tree of two-wire buses).

A second development designed for up to 250 dry electrodes on a two-wire bus (not a tree of two-wire buses) in the form of high-frequency remotely powered ASICs to enable detection of hazardous leakage current has been demonstrated. So far, the demonstration is limited to 2 sensors on the body and 15 sensors on the test bench. A next iteration to correct some implementation errors and defects is needed to address more sensors. However, the tree arrangement strategy used in the first development could also be considered as a backup solution for the ASICs, if we have difficulties to reach the goal of 250 sensors on a single two-wire bus.

In addition to the challenge of combining remote power supply, microvolt biopotential measurements, synchronization, and communication on the same two-wire bus, a significant difficulty addressed in this paper is compliance with medical standards for leakage currents in the context of wearables that can hardly provide reliable waterproof double insulation of electrical connections. Although leakage current detection was not implemented during this development, the design was made to address this issue by proposing power at frequencies where a leakage current can be detected by the central unit by monitoring the current.

## 6. Patents

The work presented in this paper is based on patent [25].

## Figures and Tables

**Figure 1 sensors-22-08219-f001:**
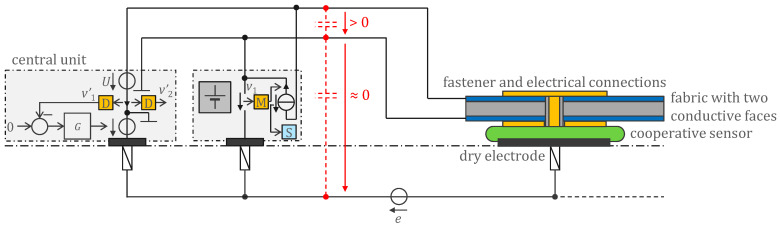
Cooperative sensors are active electrodes equipped with electronic circuits that allow them to be connected to a parallel bus of up to two wires. They are synchronized by the central unit to which they transmit their measured biopotential. For devices that do not need to be defibrillator-proof, the 2-wire bus can be made of a fabric with both sides made conductive. Insulation of the bottom side is not necessary because the voltage between the bottom side and the body is close to zero due to the *G* controller. The top side can easily be insulated with an additional layer of fabric (e.g., a regular garment), providing that excess of leakage currents is electronically detected. The small cooperative sensors are attached and connected to the fabric, making the assembly seamless while maintaining the usual properties of the fabric, i.e., flexibility, stretchability, breathability, and washability. Symbol legend in Appendix A.

**Figure 2 sensors-22-08219-f002:**
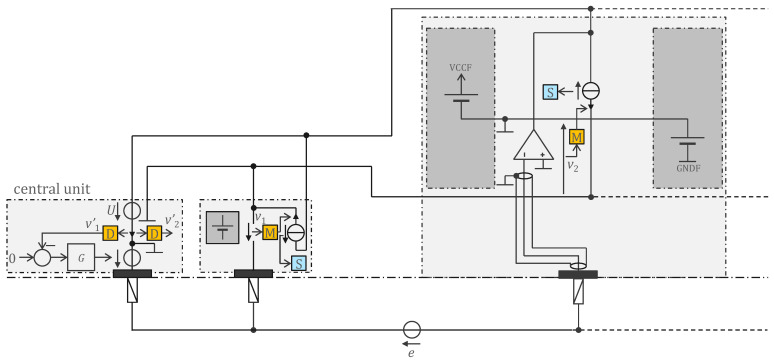
Implementation of cooperative sensors with extremely high input impedance achieved with a simple bootstrap circuit taking advantage of floating sensor batteries. The patient is protected against sensor leakage currents by diodes (not shown) that prevent stored charges from accidently leaving the sensors but allow the 2-wire bus to be used to simultaneously recharge all batteries when the system is not worn. Symbol legend in Appendix A.

**Figure 3 sensors-22-08219-f003:**
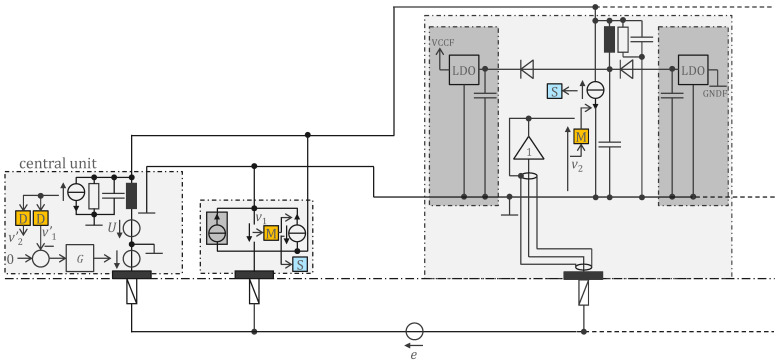
Remotely powered cooperative sensors for biopotential measurement with dry electrodes, with digital communication at 1.28 Mb/s in both directions (full duplex), and remote power supply at 500 Hz. Symbol legend in Appendix A.

**Figure 4 sensors-22-08219-f004:**
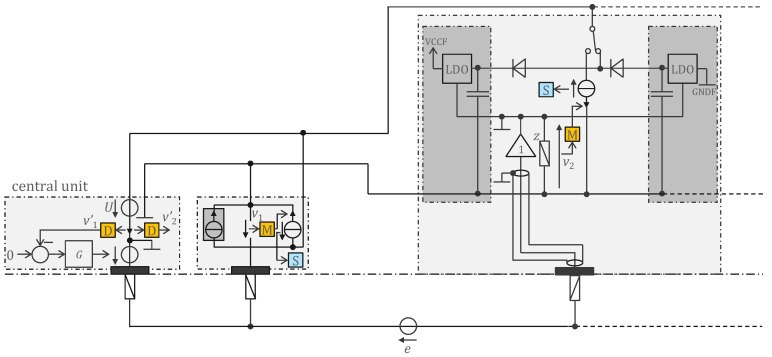
Remotely powered cooperative sensors for measuring biopotentials with dry electrodes, with digital communication at 500 000 samples per second and remote supply *U* voltage at 1 MHz. Left: schematic overview of the central unit circuit; middle: schematic overview of a sensor circuit; right: detailed circuit diagram of a sensor. Symbol legend in Appendix A.

**Figure 5 sensors-22-08219-f005:**
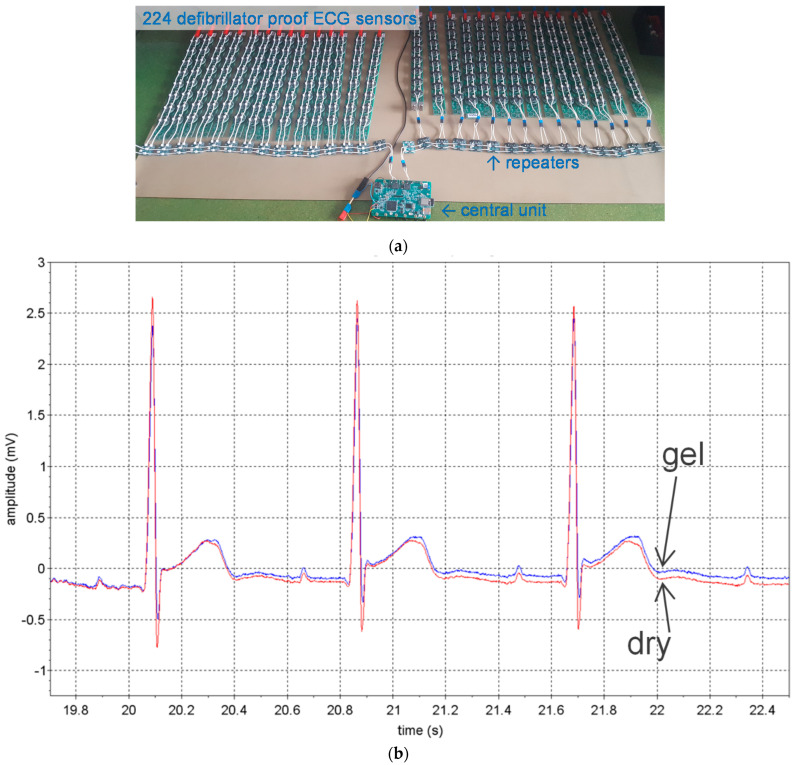
Remotely powered cooperative sensors based on the approach of Figure 3: (**a**) study prototype made of 224 sensors for ECGi with defibrillation protection between all electrodes; (**b**) onesubject trial to compare dry stainless steel electrodes with adhesive Ag/Ag^+^Cl^−^ gel electrodes showing identical noise level for both electrode types (measured from adjacent leads).

**Figure 6 sensors-22-08219-f006:**
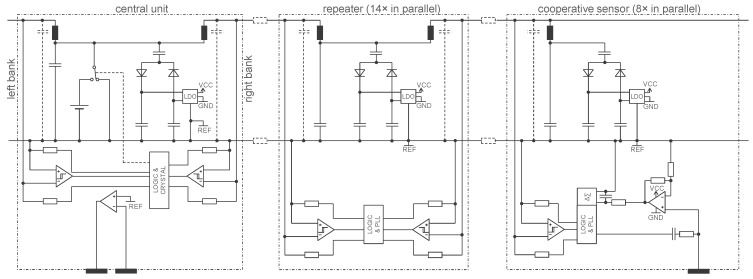
A possible implementation of the principle shown in Figure 3 (as prototyped in the device shown in Figure 5). Symbol legend in Appendix A.

**Figure 7 sensors-22-08219-f007:**
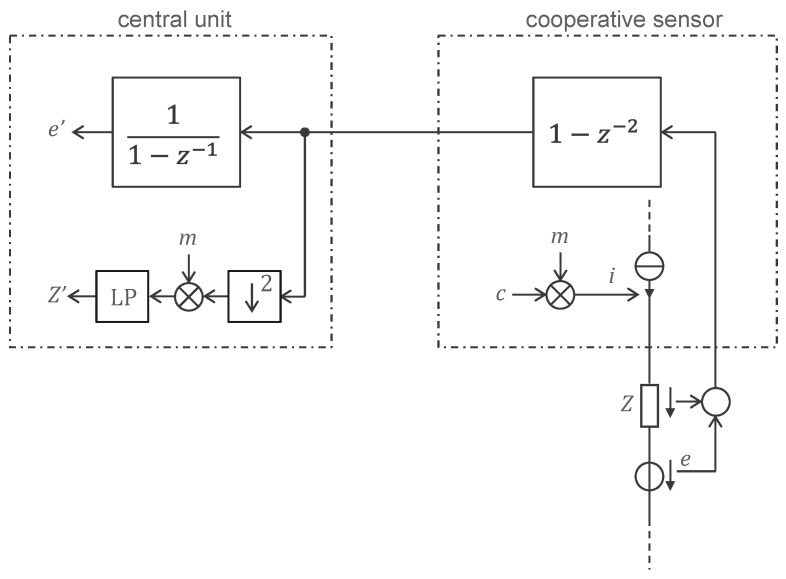
Processing scheme for the transmission of the signal from the cooperative sensor to the central unit. The sum of the biopotential *e* and the voltage drop across the skin impedance *z* is filtered by 1 − *z*^−2^ = (1 − *z*^−1^)(1 + *z*^−1^) to compress the biopotential signal (1 − *z*^−1^) and eliminate the supply disturbance at the Nyquist frequency (1 + *z*^−1^). The current *i* injected through the skin impedance *Z* is obtained from a carrier *c* (square wave at Nyquist frequency) modulated by the signal *m* (typically a square wave at 1 Hz). Decompression (lossless in the biopotential band, e.g., 0.05 to 150 Hz) with the filter 1/(1 − *z*^−1^) provides the measured biopotential *e*′ for further processing by the central unit. In parallel, the received signal is subsampled by 2 and demodulated by multiplying it by the signal *m* and filtering it with the low-pass filter LP to obtain the skin impedance signal *Z*′. Symbol legend in Appendix A.

**Figure 8 sensors-22-08219-f008:**
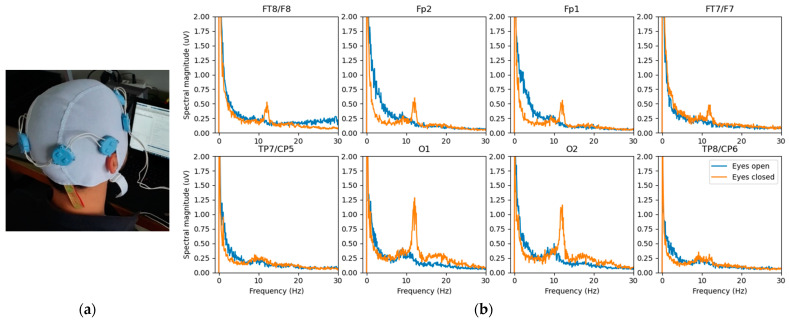
Remotely powered cooperative sensors based on the approach shown in Figure 3: (**a**) study prototype consisting of 8 EEG sensors; (**b**) power spectra for each electrode showing the increase in power at certain frequencies resulting from closed eyes (orange) versus open eyes (blue).

**Figure 9 sensors-22-08219-f009:**
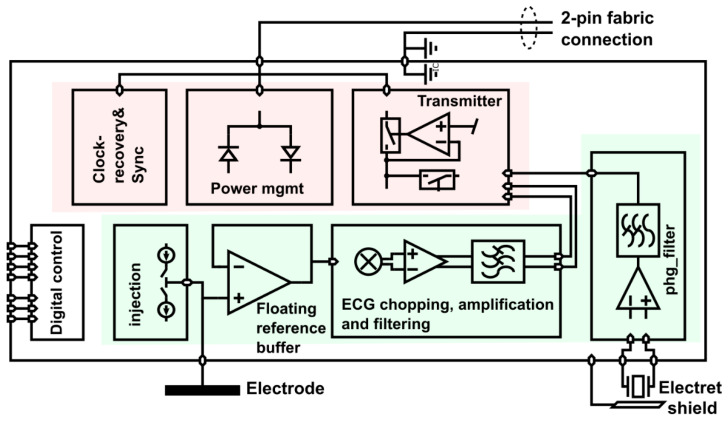
ASIC block diagram of biopotential cooperative sensor. The circuit blocks that interface with the 2-wire sensor bus are marked in red and the signal processing circuits in green. Symbol legend in Appendix A.

**Figure 10 sensors-22-08219-f010:**
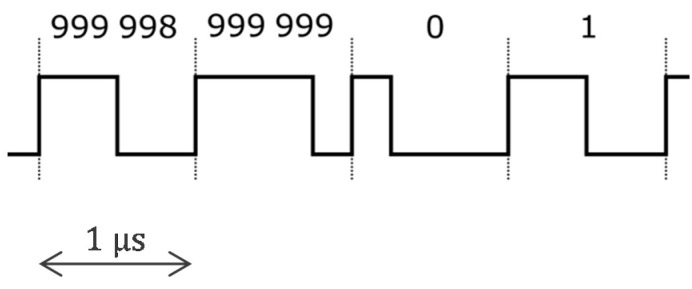
Supply voltage *U* consisting of a 1 MHz square wave with a sync marker (periodicity break) every 1 s (every 1,000,000 periods of the 1 MHz square wave).

**Figure 11 sensors-22-08219-f011:**
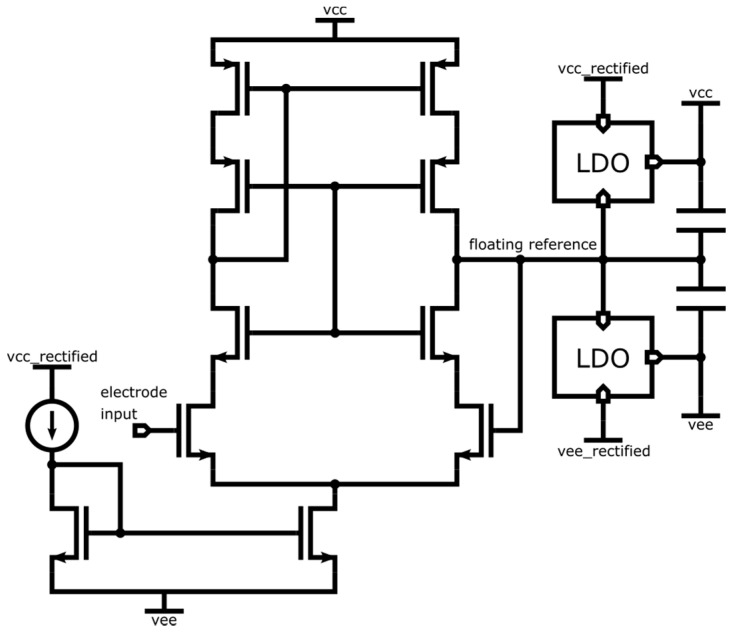
Transistor-level circuit of the floating reference buffer. The output of the unity gain buffer is used as ground for its own positive and negative supply. Symbol legend in Appendix A.

**Figure 12 sensors-22-08219-f012:**
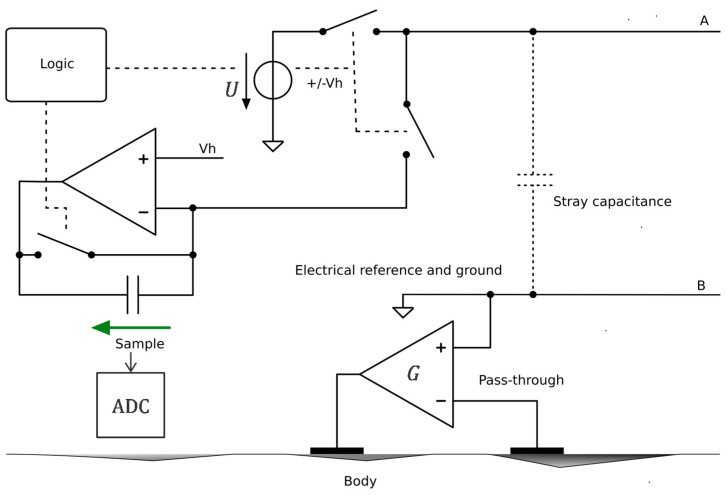
Implementation of the circuit for the central unit based on the approach shown in Figure 4. Symbol legend in Appendix A.

**Figure 13 sensors-22-08219-f013:**
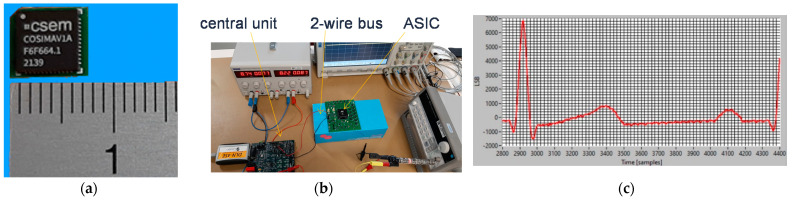
Remotely powered cooperative sensors based on the approach in Figure 4: (**a**) developed ASIC (mounted on a 7 × 7 mm^2^ micro PCB); (**b**) experimental setup during the integration process with (bottom left) the central unit and (middle) the ASIC of one sensor; (**c**) 2 mV ECG of simulator measured by a sensor.

**Figure 14 sensors-22-08219-f014:**
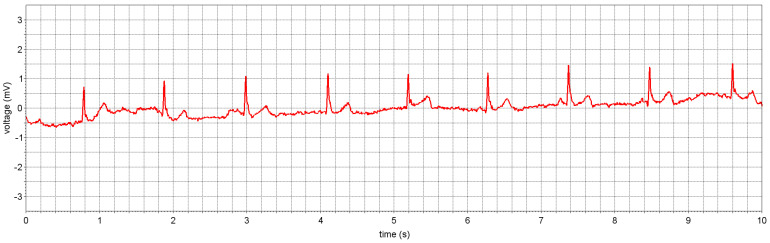
Same as Figure 13 but with two sensors applied to the body for actual ECG measurement (some 50 Hz noise remains due to limitations of the current setup).

**Table 1 sensors-22-08219-t001:** Comparison to existing work (the main contribution of the paper is highlighted in grey).

Technique/Features	[Ref], Section	Comment
Conventional star arrangement		Not suitable for wearables with many electrodes
Passive electrodes, shielded cables	[1]	Widespread
Active electrodes, two-wire cables	[2,3,20,21]	Well-known in the literature, but little used
Parallel bus arrangement		Scalable (connector size independent of nb. of electr.)
Bus with more than 2 wires	[22]	Not easily flexible, stretchable, breathable, washable
Two-wire bus (cooperative sensors)	Section 2	Simplest connection
Locally powered Bootstrapping	Section 2.1 Section 2.2	Easy to comply with safety (medical standards) Suitable for dry electrodes
Remotely powered	Section 3/Section 4	Sensors can be miniaturized
No monitoring of leakage currents No bootstrapping	Section 3.1/Section 4.1	Requires reliable waterproof double insulation Not ideal for dry electrodes
Monitorable leakage currents Bootstrapping	Section 3.2/Section 4.2	Suitably flexible, stretchable, breathable, washable Suitable for dry electrodes

**Table 2 sensors-22-08219-t002:** Acquisition chain noise (bandwidth 0.5–50 Hz).

Sensor	Peak-to-Peak Noise over 10 s (µV)	Rms Noise (µV)
1	(potential reference)	(potential reference)
2	4.01	0.59
3	4.65	0.60
4	5.18	0.66
5	4.90	0.64
6	4.88	0.66
7	5.41	0.67
8	4.40	0.66

## Data Availability

Not applicable.

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
