# Peer review of "Remotely Powered Two-Wire Cooperative Sensors for Biopotential Imaging Wearables"

_sensors, 2022, doi:10.3390/s22218219_

Round 1

Reviewer 1 Report

Authors have presented work on remotely powered 2-wire cooperative sensors for biopotential imaging wearables. Two approaches have been investigated; one is implemented with off-the-shelf components and the other with an ASIC. The experimental results show that both approaches are feasible, but the ASIC approach better addresses medical safety concerns and offers other benefits, such as lower power consumption, larger number of sensors on the two-wire bus, and smaller formfactor. Following comments would be helpful to improve the manuscript.

·         The write-up needs to be more structure and well connected.

·         Introduction is too long and highlights theory/operating techniques. It will be good to split it in to two sections i.e. Introduction, and Operational principles or Techniques.

·         Figure 9, fonts need to be larger for better readability. Some of the other figure also needs to be improved. Please check.

·         Add a conclusion section.

·         Main contributions and key findings need to be clearly highlighted. Better to include some quantitative data to highlight significance.

·         Add a table or comments in comparison to existing work to further highlight significance of the presented work.

·         Overall, the work is interesting and good, a well-connected approach is required in write-up.

Reviewer 2 Report

A very well-written article. My only comment is: VERY WELL DONE!

Reviewer 3 Report

This paper presented two cooperative sensors for medical imaging wearables for biopotentials detection. By incorporation of power function in the bus, the first sensor can work without increasing noise.  The second sensor, with ASIC approach, can potentially solve the safety issue during detection. However, the way to introduce these techniques is misleading and unprofessional. The major concerns I had for this manuscript are as following. Due to the nature of these concerns, I do not recommend its publication.

1. The introduction part is redundant. It is better to summarize all the conventional techniques with one figure or table, instead of 5 individual figures that can easily mislead the readers.
2. The sensor circuits (Figure 2, 3,4,5,7,8,) is not clear, the authors should label all the key symbols in the figures.
3. The organization and quality of figures need to be improved. There are too many figures in this manuscript. It is better to reduce the number of images by integrating data from one sensor into one graph. In addition, figures (e.g., fig.18b) should not be a screenshot, that is not professional. Furthermore, it is better to hide the body region in Figures 6a and 18a.
4. The “Materials and Methods” section is more like “ Results and Discussion”. The authors should focus on the introduction of device design and fabrication.

Round 2

Reviewer 1 Report

Authors have addressed the comments and manuscript has been improved.

Author Response

Thank you for your time for the second review.

Best regards,

Olivier Chételat

Reviewer 3 Report

This manuscript has a big improvement after revision, however, the organization of the content needs to be improved. This is not a review paper, please don't give up the "Method or Experimental section" and highlight the "State-of-the-art techniques" at length. You can use a table to compare the Cons & Pros of those techniques and leave the description/figures in the supplementary files. 

Author Response

Thank you for your time for this second review.

The titles of sections 3 and 4 have been changed to 'Method' and 'Results', as suggested. As for section 2, I understood from your comments that you think it is too long and not in the scope of the paper. Therefore, sections 2.1, 2.2, and 2.3.3 have been removed to make it as short as possible. As cooperative sensors are not known as well as the other approaches, we would like to keep sections 2.3.1 and 2.3.2 (which have been renamed in sections 2.1 and 2.2) since they are essential to understand the challenges addressed in the paper (sections 3 and  4). Table 1 that compares the pros and cons has been updated.

Round 3

Reviewer 3 Report

After careful review, I think the latest version has resolved all my concerns.